# QTc Interval Reference Values and Their (Non)-Maturational Factors in Neonates and Infants: A Systematic Review

**DOI:** 10.3390/children9111771

**Published:** 2022-11-18

**Authors:** Lisa De Smet, Nathalie Devolder, Thomas Salaets, Anne Smits, Karel Allegaert

**Affiliations:** 1Faculty of Medicine, KU Leuven, 3000 Leuven, Belgium; 2Division of Pediatric Cardiology, Department of Cardiovascular Sciences, University Hospitals Leuven, KU Leuven, 3000 Leuven, Belgium; 3Neonatal Intensive Care Unit, University Hospitals UZ Leuven, 3000 Leuven, Belgium; 4Department of Development and Regeneration, KU Leuven, 3000 Leuven, Belgium; 5Department of Pharmaceutical and Pharmacological Sciences, KU Leuven, 3000 Leuven, Belgium; 6Department of Hospital Pharmacy, Erasmus MC, 3000 GA Rotterdam, The Netherlands

**Keywords:** QTc interval, Torsades de Pointes, neonates, infants, maturational changes, pharmacovigilance

## Abstract

QTc interval measurement is a widely used screening tool to assess the risk of cardiac diseases, arrhythmias, and is a useful biomarker for pharmacovigilance. However, the interpretation of QTc is difficult in neonates due to hemodynamic maturational changes and uncertainties on reference values. To describe trends in QTc values throughout infancy (1 year of life), and to explore the impact of (non)-maturational changes and medicines exposure, a structured systematic review (PROSPERO CRD42022302296) was performed. In term neonates, a decrease was observed over the first week of life, whereafter values increased until two months of age, followed by a progressive decrease until six months. A similar pattern with longer QTc values was observed in preterms. QTc is influenced by cord clamping, hemodynamic changes, therapeutic hypothermia, illnesses and sleep, not by sex. Cisapride, domperidone and doxapram result in QTc prolongation in neonates. Further research in this age category is needed to improve primary screening practices and QTcthresholds, earlier detection of risk factors and precision pharmacovigilance.

## 1. Introduction

The neonatal electrocardiogram (ECG) has specific characteristics because of postnatal changes in hemodynamics, and uncertainties on normal and pathological values. This includes the QT interval, and its deducted version—corrected for heart rate—QTc. The QT interval is measured from the onset of the Q wave to the termination of the T wave. It represents the time from the start of ventricular depolarization to the end of repolarization [1]. QT interval prolongation is an indicator for disturbances of ventricular repolarization and elevates the risk of lethal arrhythmias, like Torsades de Pointes (TdP). 

A prolonged QT or QTc interval in neonates and infants can be acquired or congenital, or may reflect maturational findings. Causes of acquired QT or QTc interval relate to electrolyte disturbances such as hypocalcemia, -kalemia and -magnesemia, as well as to applied therapies such as (therapeutic) hypothermia or drug–drug interactions. Indeed, medicines are a common cause of acquired QT or QTc interval prolongation because these may interfere with the rapid component of the delayed rectifier potassium current—IKr—and may lead to TdP. Over the last few years, questions have arisen regarding the safety of medicines, including interest in the QTc interval of newborns and infants [2,3].

To put this into a pharmacovigilance perspective, the QT/QTc interval is used as safety marker in the development of new medicines. In phase one and two studies, QT/QTc-studies are performed to objectify QTc prolongation in healthy, young adult volunteers [4]. However, not all medicines cause QTc prolongation, and neither always elevate the arrhythmia risk. Finally, how to translate this to neonates and infants remains uncertain [3].

Besides acquired causes, congenital forms such as congenital Long QT Syndromes (LQTS) also occur. Research reported the association between LQTS and an increased incidence of sudden death. As this paper has its focus on neonates and infants, numerous hypotheses have been brought forward to link LQTS as mechanism of Sudden Infant Death Syndrome (SIDS), more recently described as Sudden Unexpected Death in Infants (SUDI). Besides precision pharmacovigilance, the development of effective screening programs in (early) neonatal life also necessitates the availability of QT and QTc reference and cut-off values. 

Since the introduction of computer-assisted automated ECG analysis, interval measurements based on ECG prints became less relevant. However, inter- and intra-observer variability still exists in the interpretation of automated measurements, as some pitfalls must be considered [2]. Often, the T-wave is incorrectly measured due to U-wave presence. This is a small wave following the T-wave and should only be included as part of the T-wave if an amplitude of ⅓ of the T-wave [5,6]. Further, to assess the QT prolongation, it must be corrected for heart rate, as QTc interval. Different mathematical formulas are used to calculate the QTc interval, with the formula of Bazett (QTcBaz), Fridericia (QTcFri), Framingham (QTcFra) or Hodges (QTcHod) [2]. Even though it is widely known that Bazett overcorrects at higher heart rates (relevant in newborns), it still is worldwide the most commonly used in neonates and young infants [7].

Unlike adults, where the reference values of the QTc interval are well described, there is no consensus on reference values in neonates and infants [2,5]. Until now, standards are commonly extrapolated from adults to neonates, despite the maturational differences. This includes the switch from right to left ventricle dominance by the first month of life, and the closure of the ductus arteriosus and foramen ovale. In addition, the decrease in pulmonary resistance causes the T-wave in V1 to deflect from positive to negative, while the increasing systemic vascular resistance results in a negative T-wave in V6 [6,8]. 

We conducted a systematic review to determine reference QT and QTc interval standards for neonates and infants, as well as the impact of its covariates, including medicines. These results can be useful for precision pharmacovigilance, to improve neonatal and infant care and to guide future research on LQTS-related SIDS prevention.

## 2. Materials and Methods

This systematic review was conducted according to the Preferred Reporting Items for Systematic Reviews and Meta-Analyses (PRISMA) guidelines. Relevant scientific articles were searched on PubMed, Embase, Web of Science (WoS) and Cochrane Library. All entries were retrieved from 15 March until 12 June 2021, supported by a librarian (Kristel Pacque, KU Leuven). Search strategies and entries are provided as Appendix A. This systematic review was registered on PROSPERO (CRD42022302296).

For pragmatic reasons (retrievability, QTc measurement tool), a restriction date of 1960 was applied when extracting articles. Articles had to be published in English, Dutch or French. The exclusion criteria were study designs including editorials and reviews, study population older than 1 year of age, outcome not matching the conditions mentioned below or unavailability of articles. Outcomes were considered appropriate if concerning QTc intervals, influenced by (1) maturational effects, (2) cofactors such as (delayed) cord clamping, hemodynamic changes, (therapeutic) hypothermia, illnesses, electrolyte disturbances, sleep position and stage and (3) medicines. Articles studying the association between LQTS and SIDS were also included if part of a boarder epidemiological study (primary screening) compared to control infants, but no full analysis on this topic was planned. The study population was considered eligible if it involved neonates or infants. To set up the upper limit of age, the standard definitions were used—from birth to 44 weeks postmenstrual age for preterm neonates and to four weeks postnatal age for term neonates, respectively—as well as for infants—up to one year of age.

Two authors (L.D.S., N.D.) constructed the set-up of search terms and entries in the four databases mentioned above, but independently performed article selection, based on title and abstract, and full text, respectively. In the event of disagreement, a consensus was found after discussion between both (L.D.S., N.D.), or a third reader (K.A.) was contacted. For each retained article, the following study characteristics were extracted: author(s), year of publication, title, study design, number and age of participants, population characteristics, timing and measurement of ECG-recording, QTc correction formula, presence of covariates and key QTc interval findings.

Risk of bias was assessed with the RoB2 tool (randomized controlled trials) [9] and ROBINS-1 tool (non-randomized controlled trials) [10]. Quality assessment of case–control, cohort and cross-sectional studies was conducted using the Newcastle Ottawa scale [11].

To further explore pharmacovigilance and its association with potential QTc effects in neonates, the association of the 100 most commonly administered medicines in the Neonatal Intensive Care Unit (NICU) with reports on longer QTc values was explored by the third reader (K.A.) using the CredibleMeds website (known TdP risk, possible TdP risk, conditional TdP risk, or medicines to avoid in congenital long QT (any of those apply) [12,13].

## 3. Results

### 3.1. Study Selection

The primary search in the above-mentioned databases resulted in 42,775 articles. After deduplication, 24,627 articles were retained (Figure 1). Primary screening based on title and abstract resulted in exclusion of 24,362 records. Of the 265 remaining articles, 62 were not retrieved. The full text of 203 articles were independently read and assessed by L.D.S. and N.D. This resulted in inclusion of 42 articles. Citation searching (snowballing) led to one extra article. In addition, experts (K.A. and Prof. Dr. Robert Ward, University of Utah, Salt Lake City, UT, USA) in this topic provided us with 28 articles from which we independently read the full text. Both experts were involved at that time in a merged Innovative Medicines Initiative (IMI)c4c and the institute for Advanced Clinical Trials (iACT)-for-children project on QTc screening in pediatric clinical trials, and both experts hereby were responsible for the neonatal aspects of this project. As part of this broader project, exchange of articles occurred, so that we were aware of some literature as provided by Prof. Dr. Robert Ward. This led to an extra inclusion of 15 articles. Of the 57 articles retained, 18 articles were on maturational effects on the QTc interval, 9 on non-pharmacological covariates, 23 on QTc prolongation due to medicines and 7 on LQTS and SIDS.

### 3.2. Quality Assessment

The 61 articles consisted of nine case–control studies, 43 cohort studies, six cross-sectional studies, two randomized controlled trials and one non-randomized controlled trial. Risk assessment of each study using Rob2 tool, ROBINS-1 tool and Newcastle Ottawa Scale, is provided in Appendix A.

### 3.3. Maturation

Concerning the evolution of the QTc interval from birth throughout infancy, 18 articles were retained for analysis. These were post hoc subdivided into early (first week) and late neonatal (week 2–4) life for analysis. Term as well as preterm neonates were included, and intervals were mainly analyzed from lead II. Values referring to QTc intervals were always measured with the Bazett formula, with occasional additional analyses, like Fridericia. The cut-off value to determine whether the QTc interval was classified as prolonged or not, varied between 440 and 470 ms. A full overview of the data extraction of these 18 articles is available in Appendix A.

#### 3.3.1. Early Neonatal Life

For the (uncorrected) QT interval, Wenger et al. [15] reported a decreasing interval with increasing age, proven by intervals of 288 and 254 (average values, no information of variability reported) ms in the first 48 h and first week, respectively. Makarov et al. [16] also observed decreasing QT intervals during the first four days. Comparing the values based on gestational age (GA), Walsh et al. [17] showed that the group of 37 neonates born preterm, had significantly shorter QT intervals compared to 68 term cases. In the most recent study of Paerregaard et al. [18], newborns [grouped by preterm (GA < 37 weeks) or term newborns (GA ≥ 37 weeks)], there was no significant difference in mean QT values (274 versus 274 ms, *p* = 0.670), respectively. Only the comparison of the two extreme groups (GA <34 to ≥42 weeks), showed shorter values in the lower GA group. The median QT for those newborns were 270 ms and 278 ms, respectively.

More data are available on the QTc interval. During the first four days of life, values fluctuate significantly. Hubscher et al. [19] observed longer QTc values on the very first day of life with mean QTc of 440 ms in newborns with a birth weight (BW) of 800–1300 g and 1800–2300 g and 420 ms in newborns of 1300–1800 g compared to the subsequent days (Appendix A). Makarov et al. [16] also reported on QTc intervals. This study showed fluctuating mean values of 434, 458 and 438 ms on the first, second and fourth day of life in term cases. As this study determined the patterns of circadian heart rhythm, they observed the heart rate dependent QT interval parameters (QT/RR slope). The first days of life are characterized by an increased QT/RR slope (which means a steep slope) compared to older children and adults. From a physiologic perspective, they hypothesized that a steep slope may reflect hyperadaptation of QT to heart rate, a flat slope hypoadaptation. Walsh et al. [20] also showed a significant shortening in maximum QTc values during the first week of life. This observation was confirmed by Schaffer et al. [21] with mean QTc values of 415 ms on the first day and 404 ms at the end of the first week. Regarding GA, no difference were observed.

On the contrary, Ulrich et al. [22] found that a higher GA was associated with a shorter QTc interval. He divided a population of 114 neonates in three cohorts from, respectively 31–34 weeks, 34–37 weeks and >37 weeks. Data showed that neonates >37 weeks had a shorter QTc interval on day one to three than preterm (31–34) neonates and a shorter interval from day two to four than preterm (34–37) neonates. Thomaidis et al. [23] confirmed this limited longer QTc interval in preterms when compared to the fifth day of life in term versus preterm neonates. He attributed this longer repolarization in preterm neonates to relative hypothermia or hypoglycaemia. However, preterm neonates are known to have higher heart rates, consequently leading to shorter RR intervals and prolonged QTc intervals. In the study of Hubscher et al. [19], with preterm neonates classified by BW, mean heart rates were 140 bpm in newborns with BW of 800–1300 g, 135 bpm in newborns with BW of 1800–2300 g.

When comparing QTc values based on ethnicity, similar patterns with decreasing QTc intervals during the first month. This was studied by Schaffer et al. [21] where 76% of the population existed of non-white infants, be it that the exact ethnical origin was not mentioned. Marti-Almor et al. [24] performed ECG during the first 48 h in 1305 newborns divided in 11 groups based on ethnicity. No significant differences were neither observed between ethnic groups. When 440 ms was taken as cut-off for prolongation, 18.33% of all newborns had a prolonged interval. This percentage was consistent in newborns of Spanish origin, Maghrebi-Near Eastern or Indian-Pakistani origin. However, when taken a cut-off of 471 ms—defined as the interval exceeding the 97.50% percentile in the largest (Spanish) group—only 4.52% of the neonates had QTc prolongation. Again, this percentage was derived from Spanish newborns and from Maghrebi and Near Eastern regions, but no statistical significance existed anymore in infants from Indian–Pakistani origin [24]. The mean QTc interval in all groups was 417.79 ms, slightly longer than reported by Schwartz [25].

#### 3.3.2. Late Neonatal Life

For QT interval values, Wenger et al. [15] showed intervals of 235 ms between week four and week six compared with 254 ms in the first week (Appendix A).

As for QTc intervals, Schwartz et al. [25] observed longer QTc intervals throughout the first six months of life (409, 406 and 400 ms at the second, fourth and sixth month), compared to the age of four days (mean QTc 398 ms) (Appendix A). Schaffer et al. [21] reported not only significantly shorter values in the first week of life, but also reported a transient significant decrease in the QTc interval during the first month of life with a mean QTc of 414 ms. A possible explanation for this discrepancy is different timing of serial measurements. After the first month, the QTc interval also slightly increased with mean values of 416 ms at two months and 417 ms at three months of age.

This subtle longer QTc interval after the first month was also seen by Rijnbeek et al. [26] with values of 419 ms (males) and 424 ms (females) between one and three months, 418 ms (males) and 422 ms (females) at three to six months, and 414 ms (males) and 411 ms (females) between six months and one year. In contrast, Uygur et al. [27] did not find this increase when they examined ECG changes by dividing 1305 children aged one day until 16 years old in ten groups according to age. Newborns from birth until one year were subdivided into five groups (0–7 d, 7–30 d, 1–3 m, 3–6 m and 6–12 m). Overall, median QTc intervals were not significantly different as values from group one to five, respectively, were 412, 411, 412, 414 and 416 ms (Appendix A, cross sectional study).

No recent study focused on the values of preterm newborns after the first month of life. Hubsher et al. [19] studied QTc values in 143 preterm newborns on the first day of life, as well as at six weeks and three months of age. He observed mean QTc values ± standard deviation (SD) of 430 ± 10 ms, 400 ± 20 ms and 400 ± 20 ms at three months in neonates with BW of 800–1300 g, 1300–1800 g and 1800–2300 g, respectively.

In 2013, Yoshinaga et al. [28] assessed the feasibility of identifying infants with LQTS at the age of one month. The mean QTc values of 4285 Japanese infants at that age was 412 ms. Five infants had QTc values exceeding 470 ms. Four of those five infants were diagnosed with LQTS. They concluded that ECG screening in infants of one month old is successful in identifying prolongation when cut-off is set to 470 ms. The current value contributes to a positive predictive value (PPV) of 80% and a negative predictive value (NPV) of 100%. Semizel et al. [29] observed the QTc values in a population of 2241 healthy Turkish Children aged one day up to 16 years and reported QTc values up to 490 ms as normal in the first six months of life. Compared to the cut-off of 470 ms determined by Yoshinaga et al. [28], a cut-off of 490 ms would result in a PPV of 100% and a NPV of 99%.

#### 3.3.3. Sex Differences in QTc Interval in Neonates and Infants

Stramba-Badiale et al. [30]. studied newborns on day three and four of age to evaluate the impact of sex. As the mean QTc intervals were 401 and 400 ms in male and female newborns, no significant differences were found between both sexes. This was confirmed by Krasemann et al. [31] in 100 neonates. In infants, there are data of Yoshinaga et al. [28] and Rijnbeek et al. [26]. The mean QTc values of 4285 Japanese infants at one month was 412 ms, with a statistical significant, but clinical very small difference between both sexes (mean QTc of 410 ms and 413 ms for males and females) [28]. This subtle longer QTc interval after the first month was also reported by Rijnbeek et al. [26,28] with values of 419 ms (males) and 424 ms (females) between one and three months, 418 ms (males) and 422 ms (females) at three to six montlhs, and 414 ms (males) and 411 ms (females) between six months and one year.

### 3.4. Non-Maturational Ovariates Influencing the QTc Interval

QTc values do not only depend on age, as covariates also affect the QTc interval. For this analysis, nine articles were retained, reporting on cord clamping (n = 1), pulmonary pressure (n = 1), therapeutic hypothermia (n = 1), illness (n = 1), electrolyte disturbances (n = 1) and sleep (n = 4). An overview of the data extraction is available in Appendix A.

#### 3.4.1. Cord Clamping

Walsh et al. [32] investigated the effect of umbilical cord clamping. A prospective cohort study included 114 term neonates, and were divided in three groups based on the timing of cord clamping: late cord clamping (three to five minutes after delivery of the feet), early cord clamping (four seconds after delivery of the feet) and cord stripping. In both early and late cord clamped neonates, the QTc interval decreases through the first week of life, but early clamped neonates showed shorter QTc values on the first day of life. Delayed cord clamping is associated with higher arterial pressure during the first hours of life, higher mean pulmonary arterial pressure (PAP) during the first nine hours of life and higher systemic pressure for 24 h along. Emmanouilidis et al. [33] reported on recordings of 28 normal newborns to investigate postnatal changes in hemodynamics on ECG parameters. Higher pulmonary arterial pressure was associated with higher P-waves and upright T-waves. So, late cord clamped neonates can be distinguished from early cord clamped neonates based on ECG differences. Based on these findings, they concluded that the volume of placental transfusion explains in part the wide range of normal values in the immediate neonatal period.

#### 3.4.2. Pulmonary Pressure

In the study of Emmanouilides et al. [33], neonates younger than one hour had significantly higher R waves in the right precordial leads and steeper S waves in the left precordial leads. During the first hour of life, mean PAP is equal to or higher than systemic pressure and a right-to-left shunt through the ductus arteriosus is usually present. Afterwards, the mean PAP declines progressively and reaches values of 50% of the native value by the end of the first day. Exploring the hypothesis that the QT interval was related to transitional physiology, no correlation between the QT interval and the degree of pulmonary hypertension or left-to-right shunt presence was observed. However, QTc values were not provided.

#### 3.4.3. Therapeutic Hypothermia

Because of the growing interest in mild therapeutic hypothermia as a neuroprotective strategy following moderate to severe hypoxic-ischaemic encephalopathy, research has been conducted on cardiovascular effects of therapeutic hypothermia. Horan et al. [34,35] investigated the relationship between body temperature and QTc values in 27 neonates, divided into five groups based on the degree and duration of cooling. Mean QTc values were measured and evolved from 431 ms at 37 °C, 459 ms at 36 °C for 24 h, 445 ms when cooled to 35 °C at 24 h, 465 ms at 34 °C for 24 h and 466 ms at 34 °C for 48 h. For each degree decrease the body temperature, the QTc increased by 3.12 ms. However, as large intra-individual variance was observed, the results could only be partially explained by temperature. Despite the significantly longer QTc time interval during therapeutic hypothermia, the absence of major cardiovascular complications is reassuring.

#### 3.4.4. Illnesses

As the QTc interval is a safety marker when medicines are administered in ill preterm and term newborns, it is important to document whether illness itself prolongs the QTc interval. A study by Shabestari et al. [36] provided data on the QTc interval in 127 healthy preterm and term neonates versus ill preterm and term neonates. The types of illnesses are provided in Appendix A (‘covariates’). The author reported that ill preterm neonates have significantly higher QTc values compared to normal preterm neonates, with values of, respectively, 418 (SD 54) ms and 386 (SD 39) ms. In this study, 11 preterm neonates died in the first 28 days of life. Those 11 cases had significantly higher QTc values than the other ill term neonates. It seems that the QTc interval is associated with mortality in preterm ill neonates, but no cut-off values were determined.

#### 3.4.5. Electrolyte Disturbances

Limited data are available on the influence of electrolyte disturbances on repolarization. Giacoia et al. [37] studied 27 term and 77 preterm neonates in the first three days of life to explore the association between total and ionized calcium and QoTc interval (i.e., the beginning of the Q-wave to the onset of the T-wave). Both in term neonates and in healthy preterm neonates, there was a significant link between total and ionized calcium levels and QoTc, with an association between the duration and interval prolongation and decreasing calcium levels. However, in the group of critically ill preterm neonates, no significant association could be detected. Giacoia assigned this to the presence of central nervous system distress in most ill neonates. The correlation improved in the absence of an intraventricular bleeding. Conversely, in the group of ill preterm neonates with hypocalcemia who received calcium gluconate infusion, a correlation was found with shortening of the QoTc interval after infusion.

#### 3.4.6. Sleep

Finally, regarding the influence of sleep on the QTc interval in newborns and infants, four articles were retrieved. Haddad et al. [38] recorded intervals in 12 term newborns during daytime sleep at two weeks of age, as well as in month one, two, three and four of life. They reported significantly longer QTc values in non-REM (Rapid Eye Movement) sleep (or quiet sleep) than in REM sleep with 439 ms and 433 ms, respectively. This difference existed in all ages. Haddad explained these results as a variability of autonomic nerve system during sleep with an increasing parasympathetic activity during non-REM sleep and an increasing sympathetic activity during REM-sleep. Not only Haddad, but also in the study of Ariagno et al. [39], intervals tended to be longer during non-REM sleep, with 443 ms at one month of age-compared to REM sleep, with 440 ms. Ariagno studied differences in prone and supine position at one and three months of age. At one month of age, values were significantly longer in prone compared to supine position. However, at three months of age this difference was no longer significant. Longer QTc intervals during prone position supported the back-to-sleep campaign to prevent SIDS. Krasemann et al. [40] confirmed the findings by Haddad and Ariagno and found significant differences in QTc values in lead II during sleep and awake periods in both boys and girls. Benatar et al. [41] reported a mean QTc interval of 416 ms during sleep although sleep stages were not subdivided into REM and quiet sleep. He observed the QT-RR relationship to be curvilinear and to be similar to the arousal state.

### 3.5. Medicines

An analysis of 27 articles was conducted to evaluate the effect of medicines on the neonatal QTc interval and risk of arrhythmias. Studies were retrieved for cisapride (n = 17), domperidone (n = 3) and doxapram (n = 2) (Appendix A). In only two of the analyzed articles, arrhythmias were detected. This is remarkably low as all the described medicines significantly prolonged the QTc interval.

#### 3.5.1. Cisapride

In two articles, QTc *shortening* was observed when cisapride was administered. The effect was observed in (non)-randomized controlled trials where the QTc interval was studied in infants treated with cisapride compared to placebo. Costalos et al. [42]. randomized 20 preterm (14 days postnatal age) infants, 10 treated with a daily dose of 0.30 mg/kg/day cisapride and 10 controls. The QTc values after seven days of treatment were significantly shorter (mean 365 ms) than in the control group (mean 393 ms). Ramìrez-Mayans et al. [43] observed 120 term infants and reported significant shortening in infants <four months. In the group treated with 0.60 mg/kg/day cisapride, 6/63 infants had a prolonged QTc interval, whereas 5/57 infants had this same effect in the control group 

Except for these two studies, the others observed longer QTc intervals. Bernardini et al. [44] performed a study verifying the potential QTc prolonging effect of cisapride. In this study, ECG recordings were made in 49 neonates before the start of treatment and at 2.9 days during treatment. Neonates were both pre- and term and were aged one to three days. A mean dose of 0.80 mg/kg/day was administered. QTc values were significantly prolonged at the second ECG recording and seven out of 49 neonates had intervals >450 ms. Six infants with prolonged intervals had a GA <33 weeks and had also lower birth weight. Khogphatthanayothin et al. [45] confirmed the above-mentioned data by finding a mean increase of QTc intervals ± SD with 15.50 ± 25 ms in 101 infants, both preterm and term. Semama et al. [46] (only term infants) and Zamora et al. [47] (preterm and term infants) both published a study in 2001 with similar findings of longer QTc intervals after starting cisapride (dose ≤ 1 mg/kg/day). Zamora et al. [48] subsequently published another study, to confirm a significant longer QTc interval in preterm infants whereas no significant differences were observed for term infants. Chhina et al. [49] also detected a longer QTc interval after cisapride administration. He noted that QTc values >441 ms on day three of recording was predictive for a prolongation >450 ms during later treatment (PPV 71%, NPV 89%). Out of 15 infants with values >450 ms, 11 had intervals >441 ms on day three.

Dubin et al. [50] evaluated the effect of the cisapride dose on QTc interval values in preterm infants. In this study, 25 preterm infants were included and subdivided into two groups (GA 31 weeks as cut-off). Both groups received a cisapride dose of 0.10 mg/kg every six hours, if necessary increased to 0.20 mg/kg every six hours. A significant longer QTc, from 410 ms to 440 ms, was observed in infants with GA <31 weeks. This was explained by the lower biotransformation capacity of cisapride by cytochrome P450 3A4. Conversely, Cools et al. [51] compared a daily dose of 0.20 mg/kg every 6 h with 0.10 mg/kg every 8 h. A significantly longer QTc time interval was observed in both groups, but lower peak values of the QTc interval were found in a three hourly compared to a six hourly regimen.

When further focusing on the effect of prematurity on prolonged intervals, both Cools and Benetar et al. [52,53] found that not prematurity itself, but postnatal age or, as Dubin et al. [50] showed, the lower biotransformation capacity of CYP450 3A4, plays a major rule in the effect of cisapride on the QT interval. Cools et al. [52] showed an increase of the QTc interval with 38 ± 20 ms in preterm infants (mean daily dose 0.80 mg/kg). Benetar et al. [53] confirmed the explanation of Dubin et al. [50] by the inverse relation between QTc and postnatal age (r = −0.06, *p* < 0.0001) for cisapride and controls.

A year later (2002), Benatar published another study with cisapride treatment in 15 preterm infants at a mean age of 24 days old. Once again, a significant longer mean QTc interval ± SD was found with values up to 454 ± 29 ms after three days, compared to 429 ± 29 ms at baseline [54]. Vandenplas et al. [55] confirmed the finding of longer values in neonates younger than three months old in a study with 150 infants treated with cisapride and 127 controls. In young (<three months of age) infants, a mean dose of 0.80 mg/kg/day was administered and significant prolonged values were observed with intervals of 447 ms in controls and 500 ms in treated infants. Corvaglia et al. [56] divided preterm infants in appropriate (AGA) and small for gestational age (SGA) and confirmed that SGA preterm infants had longer QTc values compared with AGA preterm infants. Mean QTc values ± SD at baseline and after five days were 397 ± 16 and 416 ± 34 ms for SGA and 386 ± 15 ms and 396 ± 16 ms for AGA, respectively. Kohl et al. [57] focused on birth weight when measuring QTc intervals. In this randomized controlled trial, 29 infants were treated with cisapride compared to 30 controls. Both groups were subdivided into extremely low birth weight (ELBW) and low birth weight (LBW). Significant longer QTc time intervals were documented after starting cisapride in the ELBW group, confirming the finding of a longer QTc value in lower birth weight.

Berul et al. [58] followed 36 neonates from the start of cisapride therapy to seven days later. ECG recordings were taken at different time points during this week. In addition to the Bazett formula, they also used the Fridericia formula and a study specific formula (QTcS). They found significantly longer time intervals for the QTcBaz, QTcFri and QTcS on different time points during the first day of treatment. On the second and seventh day of exposure, the QTc increased significantly. The highest mean values ± SD for QTcBaz and QTcFri were, respectively 417 ± 7.90 ms and 459 ± 7.70 ms on day seven post dose. Extensive interindividual variability was seen, and the authors concluded that the QTc values felt within the anticipated range.

#### 3.5.2. Domperidone

One study showed prolonged values during treatment. Djeddi et al. [59] allocated 31 neonates into three groups based on GA (mean dose 1.10 mg/kg/day). They confirmed a significant increase in the QTc time interval in infants with a GA of ≥32 weeks. In neonates with a GA <32 weeks, no significant prolonged values were observed. Mean overall values ± SD were 373 ± 4.87 ms before, and 387.20 ± 5.10 ms after 2.50 ± 1.50 days of treatment. Two other studies could not confirm these results. Günlemez et al. [60] performed a prospective study of 40 premature infants and started treatment with 1 mg/kg/day domperidone at 32 days postnatal age. Mean baseline values were 370 ms. After three, five and seven days of treatment, the mean intervals were 380 ms, 370 ms and 370 ms, respectively. Two infants had prolonged intervals (>450 ms), which normalized after discontinuation. Vieira et al. [61] studied preterm and term infants at mean age of 26 days (0.50 to 1 mg/kg/dose, three-four times per day). No significant longer QTc values were found as baseline values were 390 ms and values collected seven to 14 days after starting treatment were 397 ms (*p* = 0.13).

#### 3.5.3. Doxapram

Miyata et al. [62] conducted a prospective cohort study in 15 preterm (mean GA 30 weeks) infants. ECG recordings were collected before and 24 h after doxapram administration (0.20 mg/kg/h). Although significant changes in mean QTc interval ± SD were observed (408 ± 48 ms to 418 ± 30 ms), all values were within the physiologically acceptable range. None of them had QTc intervals exceeding 440 ms or experienced cardiac arrhythmias. Maillard et al. [63] performed a similar study (45 preterm infants, mean GA 28.90 weeks) during continuous intravenous doxapram (0.50–1 mg/kg/h). A moderate but statistically significant longer (*p* = 0.0065) QTc interval was observed (394 ± 4 ms before, and 409 ± 4 ms at 48 and 72 h after treatment initiation). In six infants, QTc values exceeded 440 ms, but no arrhythmias were detected.

#### 3.5.4. Medicine Utilization in the NICU, and Its Association with Potential and Reported QTc Effects

The medicines listed in the most recent top 100 on medicine utilization were screened on the likelihood of QTc prolongation, using the CredibleMeds website [known risk of TdP, possible risk of TdP, conditional risk of TdP or medicines to avoid in congenital long QT (any of those apply)]. Of 24 out of the top 100 medicines, evidence was available of one of the four categories mentioned. The medicines retained, and their ranking in the top 100 are provided in Table 1.

### 3.6. Long QT Syndrome

Related to the assocation between long QT syndrome and SIDS, Kelly et al. [64] studied 21 near-miss SIDS infants from whom three died compared to 45 control infants and 861 random infants from the normal population. No significant prolonged QTc intervals were observed in the near-miss cases as mean QTc intervals were 390 ms in both near-miss and controls. Kelly et al. [64] mentioned that no defibrillation was needed in any of the near-miss SIDS infants, whereas this would be the case if long QT syndrome would be a main cause of SIDS. Montague et al. [65] observed similar findings, studying 17 infants who were at risk for SIDS—with positive family history or for investigation with unexplained apnea—and 17 controls. Southall et al. [66] neither observed a significant trend to longer QTc values, when comparing 15 SIDS cases with age-matched controls.

Furthermore, Haddad et al. [67] found even significantly shorter QTc values in near-miss SIDS infants than observed in controls, at the age of three to four months. Recordings were made in seven near-miss SIDS cases one week until four months after the event. These results were compared to 12 controls. A similar pattern with shorter QTC values in cases was observed in the study of Weinstein et al. [68].

Conversely, Schwartz et al. [69] showed in a prospective study significant longer QTc intervals in infants who subsequently died from SIDS. Recordings were made in 34,442 term infants on the third or fourth day of life and were followed up for one year to assess SIDS occurrence. After one year, 24 infants died of SIDS with a mean QTc of 435 ms in this population. This was significantly longer than the mean QTc of 400 ms in survivors and 392 ms in infants who died from other causes, but does not mean that all these cases had LQTS.

Regarding the etiology of channelopathies such as the Congenital Long QT Syndromes itself, 75% of LQTS cases are linked to mutations in genes encoding for voltage-gated potassium channel subunits (KCNQ1, KCNH2, KCNE1, KCNE2), or for voltage-gated sodium channel SCN5A. The cohort study of Millat et al. [70] screened 52 cases, who died unexpectedly before 12 months of age, for mutations in the latter genes. LQTS mutations were detected in five of the 32 identified SIDS cases. In only two of them, no additional risk factors were identified.

## 4. Discussion

This systematic review reflects the difficulty in interpreting the neonatal QTc interval, as the wide variation of values throughout infancy was confirmed. We have provided all values as reported in the specific papers in the Appendix A to ensure full and easy accessibility.

Interpretation of the QTc value is more straightforward in adults as reference values are widely studied and available. Cut-off QTc interval values according to the European Society of Cardiology guidelines are 450 ms and 470 ms for adult males and females [71]. This discrepancy in QTc values between males and females is less observed in newborns as female newborns have shorter QT values compensated by shorter cycle length resulting in similar QTc values in females and males during this period of life [28,29]. Although sex differences in this population are described in ECG—in the R-wave in V6 and the S-wave in the left precordial leads for example—no significant differences were found for the QTc interval [28]. Marcellino reported a statistical difference, but clinical not relevant difference between male to female newborns on the QTc interval (398, SD 29 versus 397, SD 33 ms, +1 ms in female newborns) [72] The same applies for ethnicity, whereas a small tendency towards longer intervals could be observed in Caucasian adults compared to people from African American origin [73].

When we focus on early neonatal life, a progressive decrease over the first days or week of life has been repeated observed. Related to this, Schwartz recommended to perform the first ECG recordings in newborns after the third day of life if used for screening and to improve accuracy and avoid variability [69]. Paerregaard et al. [18] showed in 2021 rather stable values in the first two weeks (mean QTc interval values of 413 and 416 ms), whereafter he confirmed a slight increase with values of 432 ms at the age of one month, with a subsequent progressive decease. When focusing on GA, shorter QT intervals were found in preterm infants compared to term newborns. The opposite effect was observed for QTc intervals, most likely due to increased heart rate in preterm infants which causes prolonged intervals by shortening the heart rate interval (RR) [17,18]. In contrast to these differences between preterm and term cases seen at birth, the patterns observed suggests that QTc in preterms are comparable to term infants from three months onwards.

This systematic review further structured the information on several covariates affecting the QTc interval. This includes covariates such as late cord clamping, therapeutic hypothermia, hypocalcemia, illness, non-REM sleep and prone sleeping position. Horan et al. [34,35] measured a temperature dependent QTc time interval in 27 neonates undergoing therapeutic hypothermia while receiving extracorporeal membrane oxygenation (ECMO). A more recent meta-analysis on the effect of therapeutic hypothermia on QTc, confirmed this finding and added that these observations normalize immediately afterwards. In the latter study, an increase of QTc time of 28 ms per decreasing degree in body temperature was observed. We must take into account that this value is much higher than reported by Horan et al. [34,35] A possible explanation therefore is that ECMO itself affects hemodynamics resulting in a more restricted increase of 3.12 ms per degree (°C) decrease.

As calcium is one of the major electrolytes contributing to the plateau phase of the action potential, Giacoia et al. [37] observed longer QoTc values with decreasing calcium levels in term and healthy preterm neonates, but not in ill preterm neonates. The pathway, responsible for longer QoTc time intervals may relate to dysfunction of cardiac sympathetics or variations in plasma catecholamine levels. More research is needed to explore how low calcium levels affect the interval in preterm ill neonates.

As QTc intervals are used for pharmacovigilance in neonates to screen for the risk on TdP, an up-to-date analysis regarding the possible side effects of medicines is important. Most evidence according to QTc prolonging medicines was available on cisapride, domperidone and doxapram. This can be explained by the fact that these medicines are not only in newborns and infants, but also in other populations widely studied for their effect on QTc prolongation.

Since the withdrawal of cisapride from the market, domperidone has taken its place in the treatment of gastroesophageal reflux. As domperidone has the same inhibiting effect on rapid outward potassium current channels (IKr), prolonged QTc intervals were not only confirmed in cisapride, but also when treated with domperidone [60,74]. Of the three articles analysed in this systematic review, only Djeddi et al. [59] showed significant longer QTc in infants >32 weeks GA. Next to cisapride [42,43,44,45,46,47,48,49,50,51,52,53,54,55,56,57,58,75,76] and domperidone [59,60,61,74,76], doxapram has also been studied because of its presumably presence of adverse cardiac effects such as second-degree AV-block and longer QTc intervals. Doxapram hydrochloride is a powerful respiratory stimulant used for idiopathic apnea of prematurity unresponsive to methylxanthines. In both studies, a slight significant increase in QTc values was observed, however no arrhythmias were witnessed [62,63].

Unfortunately, it is not because most studies were focusing on the four medicines analyzed above, that other medicines are safe and free of the risk of QTc prolongation. In 24 out of the top 100 used medicines in a NICU setting—from a study set up by Stark et al. [12,13]—an association with a potential risk for TdPare suggested in the credibleMed website, be it that these signals have not been reported or detected in neonates. This means that much more research will be necessary to reveal more accurate data on these possible adverse effects in this specific population [62]. As part of such a research plan, we should also be aware that medicines are not only potentially dangerous when directly administered to neonates or infants, but there also exists a risk in an indirect way, more specific by passing the placenta or the blood-milk barrier during lactation. The relevance of these indirect exposure on the QTc intervals in early neonatal life have been illustrated for selective serotonin-reuptake inhibitors, methadone and hydroxychloroquine [77,78,79].

Finally, congenital forms of QTc prolongation are also well described. The Long QT Syndrome is a channelopathy characterized by prolonged QTc intervals, thereby elevating the risk of Torsades de Pointes. TdP is a polymorphic ventricular tachycardia and can lead to ventricular fibrillation and sudden cardiac death. The vast majority of LQTS is caused by a mutation in either voltage-gated potassium channel subunits, or a voltage-gated sodium channel and leads to a prolonged cardiac action potential. LQTS has a prevalence of 1 in 2000 infants [5].

Regarding LQTS and its association with an increased risk of sudden death in infants, there have been many speculations circulating. Sudden Infant Death Syndrome is a multifactorial disorder defined as a sudden death of an infant that is unexpected by history and in which postmortem examination fails to demonstrate a cause of death [70,80,81]. Since several risk factors for SIDS were identified throughout the years (prone sleeping position, temperature higher than 20 degrees in the bedroom, inside smoking), measures were taken to reduce the latter. Despite this reduction, SIDS remains one of the largest single causes of death between one week and one year of age [70,80,81].

Although the structured approach and summary of the available data is a strength, this systematic review has also some limitations. A date and language restriction was used in this review and did therefore not contain all the appropriate studies. Although the risk of bias assessment was assessad as low in most of the studies, one must be aware that the main part of the included study designs consisted of observational studies which inherently carry a higher risk of bias (level four of evidence). Furthermore, reported data were sometimes incomplete and raw data were commonly not available. Some studies performed one serial recording and did not complete a period of follow-up, which did not allow us to evaluate the progression of individual QTc values during the first year of life. Finally, there was no standardized data collection. Because articles were published over a period of 60 years, much technological developments occurred. Consequently, heterogeneity was present concerning study population, in- and exclusion criteria and the moment of ECG recording in neonates and infants such as sleep, during feeding, day- and nighttime.

At present, it remains difficult to come up with robust QT_C_ values as reference values or thresholds based on the current systematic assessment of the literature. We have provided all QTc values as reported in the Appendix A). As we do need such reference values for both precision pharmacovigilance in neonates and infants, as well as to construct potential cardiac screening programs, a next obvious step would be to pool individual QT and QTc observations, preferably based on longitudinal studies, and combined with the relevant maturational and non-maturational covariates identified with this detailed, structured assessment of the available literature. Until then, we suggest considering to use the most recently reported QTc values in the large database of Paerregaard, be it that this mainly includes caucasian newborns [18].

## 5. Conclusions

We provide a systematic overview on reference QTc intervals during the first year of life, its maturational and non-maturational covariates, including medicines that influence this interval. The identification of these covariates implies necessary precautions, screening or mitigation strategies. However, a significant longer QTc time interval is not inherently associated with an increased risk for cardiac arrhythmias, and the list of reported medicines that prolong QTc is still limited. Related to LQTS screening, there is not yet consistency on its relevance and timing. Further research holds the promise to improve primary screening, earlier detection of risk factors and precision pharmacovigilance.

## Figures and Tables

**Figure 1 children-09-01771-f001:**
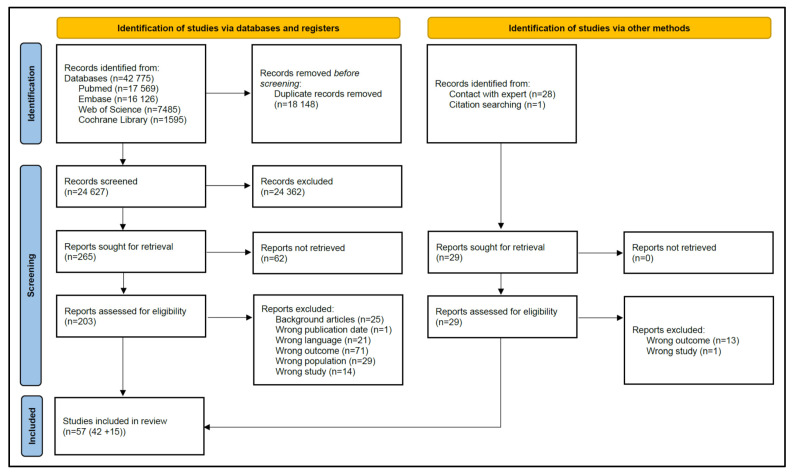
PRISMA 2020 flow diagram [14].

**Table 1 children-09-01771-t001:** Assessment of potential risk for Torsades de Pointes using the CredibleMeds website from a list of 100 medicines administered in Neonatal Intensive Care Unit-setting listed by Stark et al. [12,13].

7. Furosemide	30. Epinephrine	51. Cotrimazole	67. Omeprazole
11. Dopamine	36. Dobutamine	53. Phenylephrine	71. Amphotericin B deoxycholate
15. Fluconazole	37. Chlorothiazide	57. Epinephrine–racemic	73. Hydrochlorothiazide
16. Erythromycin	40. Metronidazole	58. Famotidine	88. Azithromycin
21. Albuterol	41. Lansoprazole	63. Metoclopramide	90. Dexmedetomidine
29. Piperacillin–tazobactam	48. Methadone	65. Levetiracetam	93. Chloral hydrate

## Data Availability

Not applicable.

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
