# Peer review of "QTc Interval Reference Values and Their (Non)-Maturational Factors in Neonates and Infants: A Systematic Review"

_children, 2022, doi:10.3390/children9111771_

Round 1

Reviewer 1 Report

The authors performed a systematic review on an important if still debated topic in pediatric cardiology: the QTc. The research is well-executed and tries to focus on the one hand on the physiological maturation of the heart in the first month and year of life and on the other hand on possible factors (drugs first and foremost) that could lead to QT prolongation. However, in both the abstract and the discussion the authors should include some more data on the actual Qtc intervals that emerged as reliable from the systematic review.

page 3 line 99: These articles do not completely respect the aims of the study, so probabily should not be considered

page 3 line 131: In methods it is not specified this: how were these experts contacted and why them in particular?

page 11 line 498: This statement should be supported by more evidences; for example, Marcellino et al. reported longer QTc values in females

page 11 line 508: The authors' rigorous review shows that the results in the literature are often conflicting; however, net results are reported in discussion; therefore, it should be specified on what basis: bias, cohort number, quality of study?

Author Response

Reviewer 1

The authors performed a systematic review on an important if still debated topic in pediatric cardiology: the QTc. The research is well-executed and tries to focus on the one hand on the physiological maturation of the heart in the first month and year of life and on the other hand on possible factors (drugs first and foremost) that could lead to QT prolongation. However, in both the abstract and the discussion the authors should include some more data on the actual Qtc intervals that emerged as reliable from the systematic review.

We thank the reviewer for the overall very positive and supportive assessment of the paper. We understand the reflection on the need for more data on the actual QTc intervals very well to make the paper of further practical use. However, and as to a large extent also reflected in your comment lower (pg 11, line 498), there is quite some variability in QTc thresholds and reference values when we assessed all currently reported literature, so that it is quite difficult to come up with generally well accepted and supported values.

Until a full individual data analysis would be done, we ‘feel’ – as this is somewhat not a fully objective assessment - that the latest paper of Paeregaard et al is likely a very reliable start for reference values of QTc time in newborns, be it that this is mainly a Caucasian cohort. We have adapted the text on this aspect, be it with these limitations added (discussion part, last alinea).

page 3 line 99: These articles do not completely respect the aims of the study, so probabily should not be considered

Sorry that we were not sufficiently clear on this. As long as these studies were ‘epidemiologic’, i.e. primary screening in any newborn, and not targeted, these data do contribute to the knowledge on reference values, so that we have added this to the paper. This was not based on eg SIDS or near-SIDS events. A reflection of this is eg the Marcellino paper, mentioned lower by the reviewer (primary screening on long QTc) to be relevant.

page 3 line 131: In methods it is not specified this: how were these experts contacted and why them in particular?

At that time, we (both experts mentioned) were collaborating between IMIc4c and iACT on a broader assessment on how to use QTc times in pediatric life during clinical trials. This effort was not restricted to newborns and infants, but myself (karel Allegaert) and prof R Ward were responsible for the neonatal/infant section. During this task, there was quite some active exchange of documents (papers, abstracts or textbook chapters) so that we had these documents. As these were provided by prof R Ward, we felt that we had to describe this in the methods section. As a reflection of this collaboration, we refer to the recently published QTc paper during and after therapeutic hypothermia in neonates (Allegaert et al, Children 2021, PMID 34943349).

We have further adapted this text section.

page 11 line 498: This statement should be supported by more evidences; for example, Marcellino et al. reported longer QTc values in females

This is indeed correct, we have added this to the discussion section of the paper, be it that this difference (absolute value 1 msec) is rather of statistical significance, and cannot really be used in the clinical setting (398, SD 29 versus 397, SD 33 msec).

page 11 line 508: The authors' rigorous review shows that the results in the literature are often conflicting; however, net results are reported in discussion; therefore, it should be specified on what basis: bias, cohort number, quality of study?

Reviewer 2 Report

This is an extensive study that analyzes an important aspect of the characteristics of the QT interval in childhood and the factors that must be assessed.

It is an analysis with an appropriate level for the recipients of this magazine that can help them make decisions in this regard.

Author Response

Reviewer 2

This is an extensive study that analyzes an important aspect of the characteristics of the QT interval in childhood and the factors that must be assessed.

It is an analysis with an appropriate level for the recipients of this magazine that can help them make decisions in this regard.

We thank the reviewer 2 for the very supportive assessment, both in wording, as well as based on classifications used in the assessment tool (review report form).
